# TAMING THE LONG TAIL OF DEEP PROBABILISTIC FORECASTING

## ABSTRACT

Deep probabilistic forecasting is gaining attention in numerous applications from weather prognosis, through electricity consumption estimation, to autonomous vehicle trajectory prediction. However, existing approaches focus on improvements on average metrics without addressing the long tailed distribution of errors. In this work, we observe long tail behavior in the error distribution of state-of-the-art deep learning methods for probabilistic forecasting. We present two loss augmentation methods to reduce tailedness: Pareto Loss and Kurtosis Loss. Both methods are related to the concept of moments, which measures the shape of a distribution. Kurtosis Loss is based on a symmetric measure, the fourth moment. Pareto Loss is based on an asymmetric measure of right tailedness and models loss using a Generalized Pareto Distribution (GPD). We demonstrate the performance of our methods on several real-world datasets, including time series and spatiotemporal trajectories, achieving significant improvements on tail error metrics, while maintaining and even improving upon average error metrics.

## 1 INTRODUCTION

Probabilistic forecasting is one of the most fundamental problems in time series and spatiotemporal data analysis, with broad applications in energy, finance, and transportation. Deep learning models Li et al. (2019); Salinas et al. (2020); Rasul et al. (2021a) have emerged as state-of-the-art approaches for forecasting rich time series and spatiotemporal data with uncertainty. In several forecast competitions, such as the M5 forecasting competition Makridakis et al. (2020), Argoverse motion forecasting challenge Chang et al. (2019), and IARAI Traffic4cast contest Kreil et al. (2020), almost all the winning solutions are based on deep neural networks.

Despite encouraging progress, we observe that *the forecasting error for deep learning models has long-tail behavior*. This means that a significant amount of samples are very difficult to forecast. These samples have errors much larger than the average. Figure 1 visualizes an example of long-tail behavior for a motion forecasting task. Existing works often measure forecasting performance by averaging across test samples. However, average performance measured by metrics such as root mean square error (RMSE) or mean absolute error (MAE) can be misleading. A low RMSE or MAE may indicate good average performance, but it does not prevent the model from behaving disastrously in critical scenarios.

From a practical perspective, the long-tail behavior in forecasting error is alarming. In motion forecasting, the long tail could correspond to crucial events in driving, such as turning maneuver and sudden stops. Failure to accurately forecast in these scenarios would pose paramount safety risks in route planning. In electricity forecasting, these high errors could be during short circuits, power outages,

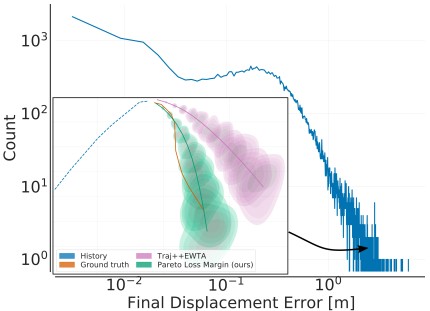

Figure 1: Log-log error distribution plot for trajectory prediction on the ETH-UCY dataset using SoTA (Traj++EWTA). We see the long tail in error upto 2 orders of magnitude higher than the average. Also shown is a tail sample with predictions from our method(teal) and Traj++EWTA(purple).

grid failures, or sudden behavior changes. Focusing solely on average performance would ignore the electric load anomalies, significantly increasing maintenance and operational costs.

Long-tailed learning is heavily studied in classification settings, with a focus on class imbalance. There is also rich literature for heavy-tailed time series Kulik & Soulier (2020). However, long tail there usually refers to *distribution of the data*, not *distribution of the error*. We refer the reader to Table 2 in Menon et al. (2020) and the survey paper Zhang et al. (2021) for a complete review. Most common approaches to address the long-tail data distribution include post-hoc normalization Pan et al. (2021), data resampling Chawla et al. (2002); Torgo et al. (2013), loss engineering Lin et al. (2017); Lu et al. (2018), and learning class-agnostic representations Tiong et al. (2021). These approaches implicitly assume strong correspondence between data and error. Hence, they are not directly applicable to forecasting, as we do not have pre-defined classes or the prediction error before training. Makansi et al. (2021) observed similar long-tail error in trajectory and proposed to use Kalman filter prediction performance to measure sample difficulty. However, Kalman filter is a different model class and its difficulties do not translate to deep neural networks used for forecasting.

In this paper, we address the long-tail behavior in prediction error for deep probabilistic forecasting. We present two loss augmentation methods: Pareto Loss and Kurtosis Loss. Kurtosis Loss is based on a symmetric measure of tailedness as a scaled fourth moment of a distribution. Pareto Loss uses the Generalized Pareto Distribution (GPD) to fit the long-tailed error distribution. The GPD can be described as a weighted summation of shifted moments, which is an asymmetric measure of tailedness. We investigate these measurements as loss regularization and reweighting approaches for probabilistic forecasting tasks. We achieve significantly improved tail performance compared to the base model and baselines. Interestingly, we also observe better average performance in most settings.

In summary, our contributions are

- We identify long-tail behavior in forecasting error for deep probabilistic models.
- We investigate principled approaches to address this long-tail behavior and propose two novel methods: Pareto Loss and Kurtosis Loss.
- We significantly improve the tail errors on four real world forecasting tasks, including two time series and two spatiotemporal trajectory forecasting datasets.

## 2 RELATED WORK

**Deep probabilistic forecasting.** There is a flurry of work on probabilistic forecasting using deep neural networks. A common practice is to combine classic time series models with deep learning, resulting in DeepAR Salinas et al. (2020), Deep State Space Rangapuram et al. (2018), Deep Factors Wang et al. (2019) and normalizing Kalman Filter de Bézenac et al. (2020). Others introduce normalizing flow Rasul et al. (2021b), denoising diffusion Rasul et al. (2021a) and particle filter Pal et al. (2021) to deep learning. For probabilistic trajectory forecasting, a few recent works propose to approximate the conditional distribution of future trajectories given the past with explicit parameterization Tang & Salakhutdinov (2019); Luo et al. (2020), CVAE Sohn et al. (2015); Lee et al. (2017); Salzmann et al. (2020) or implicit models such as GAN Gupta et al. (2018); Liu et al. (2019a). Nevertheless, most existing works focus on average performance, the issue of long-tail in error distribution is largely overlooked in the community.

**Long-tailed learning.** The main efforts to address the long-tail in error in learning revolve around reweighing, resampling, loss function engineering, and two-stage training, but mainly for classification. Rebalancing during training is done in the form of synthetic minority oversampling Chawla et al. (2002), oversampling with adversarial examples Kozerawski et al. (2020), inverse class frequency balancing Liu et al. (2019b), balancing using effective number of samples Cui et al. (2019), or balance-oriented mixup augmentation Xu et al. (2021). Another direction involves post-processing either in form of normalized calibration Pan et al. (2021) or logit adjustment Menon et al. (2020). An important direction is loss modification approaches such as Focal Loss Lin et al. (2017), Shrinkage Loss Lu et al. (2018), and Balanced Meta-Softmax Ren et al. (2020). Others use two-stage training Liu et al. (2019b); Cao et al. (2019) or separate expert networks Zhou et al. (2020); Li et al. (2020); Wang et al. (2021). We refer the readers to Zhang et al. (2021) for an extensive survey. Tang et al. (2020) indicated SGD momentum can contribute to the aggravation of the long-tail problem and suggested de-confounded training to mitigate its effects. Feldman (2020); Feldman & Zhang (2020) performed theoretical analysis and suggested label memorization in a long-tail distribution as a necessity for the network to generalize.

A few methods were developed for imbalanced regression. Many approaches are modifications of SMOTE (Synthetic Minority Oversampling Technique) such as, adapted to regression SMOTER Torgo et al. (2013), augmented with Gaussian Noise SMOGN Branco et al. (2017), or Ribeiro & Moniz (2020) extending for prediction of extremely rare values. Steininger et al. (2021) proposed DenseWeight, a method based on Kernel Density Estimation for better assessment of the relevance function for sample reweighing. Yang et al. (2021) proposed a distribution smoothing over label (LDS) and feature space (FDS) for imbalanced regression. Prasad et al. (2018); Zhu & Zhou (2021) worked on robust regression approaches applicable to point forecast. GARCH Bollerslev (1986) and AFTER Cheng et al. (2015) addressed heavy-tailed error in forecasting but both are statistical models, and not applicable to deep learning. A concurrent work is Makansi et al. (2021) where they also notice the long-tail error distribution for trajectory prediction. They use Kalman filter Kalman (1960) performance as a difficulty measure and propose contrastive learning to mitigate the tail problem. However, the tail samples of Kalman Filter differ from that of deep learning models.

Most methods in long-tailed learning operate on *known heavy-tailedness* in data, whereas our focus is to mitigate the unknown long tail in the error distribution of test samples without any specific assumption on the data distribution. This is essential to our problem setting and techniques.

## 3 METHODOLOGY

We first identify the long-tail error distribution in probabilistic forecasting. Then, we propose two novel methods, Pareto Loss and Kurtosis Loss, to mitigate the long tail in error.

### 3.1 LONG-TAIL IN PROBABILISTIC FORECASTING

Given input $x_t \in \mathbb{R}^{d_{in}}$ and output $y_t \in \mathbb{R}^{d_{out}}$ respectively, probabilistic forecasting task aims to predict the conditional distribution of future states $\mathbf{y} = (y_{t+1}, \ldots, y_{t+h})$ given current and past observations $\mathbf{x} = (x_{t-k}, \ldots, x_t)$ as:

$$p(y_{t+1}, \ldots, y_{t+h} | x_{t-k}, \ldots, , x_t) \tag{1}$$

where $k$ is the length of the history and $h$ is the prediction horizon. The maximum likelihood prediction –mean when the predicted distribution is a Gaussian– can be denoted as $\hat{\mathbf{y}} = (\hat{y}_{t+1}, \ldots, \hat{y}_{t+h})$.

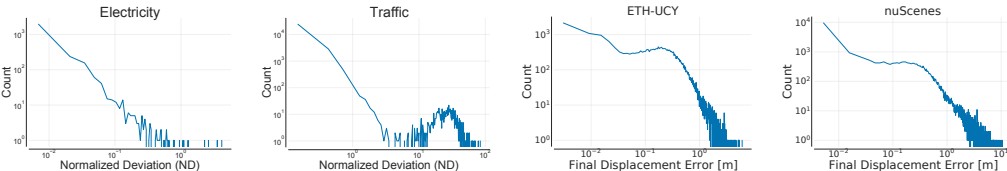

Figure 2: Log-log error distribution plots. Time series datasets (left half) use DeepAR, trajectory datasets (right half) use Traj++EWTA. This clearly illustrates the long tail in error distribution.

Long tailed error distributions for deep learning models manifest in numerous real world datasets. This is evident in four benchmark forecasting datasets studied in this work (Time series: Electricity Dua & Graff (2017), Traffic Dua & Graff (2017); Trajectory: ETH-UCY Pellegrini et al. (2009); Lerner et al. (2007), nuScenes Caesar et al. (2020)). Fig. 2 shows the long-tailed error distribution for time series datasets using DeepAR Salinas et al. (2020) and for trajectory datasets using Trajectron++EWTA Makansi et al. (2019). We follow the literature and use Normalized deviation (ND) and Final Displacement Error (FDE) to measure the performance.

We also observe that the samples forming the tail in error vary across methods and even across different runs of the same model. For example, we trained 2 DeepAR Salinas et al. (2020) models on the same Electricity forecasting dataset from UCI repository Dua & Graff (2017). We observe that the sets of samples with the top 5% error values have only 3.5% samples common to both models. This shows that the tail in the data does not necessarily correspond to the tail in error.

The fact that it is impossible to identify a fixed set of tail samples means that we cannot simply reweigh ( Cui et al. (2019); Fan et al. (2017)) or resample ( Torgo et al. (2013); Branco et al. (2017)) these samples before training. The variation of tail samples between models also invalidates the approach taken by Makansi et al. (2021). Mitigating the long tail in error requires an approach that is independent of the data distribution and is adaptive during training. Thus, we propose using tail-sensitive loss augmentations that adapt the model to also improve on samples with tail errors.

## 3.2 Pareto Loss

Long tail distributions naturally lend themselves to analysis using Extreme Value Theory (EVT). EVT McNeil (1997) shows that long tail behavior of a distribution can be modeled as a generalized Pareto distribution (GPD). The probability distribution function (pdf) of the GPD is:

$$f_{(\xi,\eta,\mu)}(a) = \frac{1}{\eta}\left(1 + \xi\left(\frac{a-\mu}{\eta}\right)\right)^{-(\frac{1}{\xi}+1)} \quad \Rightarrow \quad f_{(\xi,\eta)}(a) = \left(1 + \frac{\xi a}{\eta}\right)^{-(\frac{1}{\xi}+1)} \tag{2}$$

where the parameters are location ($\mu$), scale ($\eta$) and shape ($\xi$). Without loss of generality, $\mu$ can be set to 0. We can drop the scaling term $\frac{1}{\eta}$ as the pdf will be scaled using a hyperparameter.

The idea behind our Pareto Loss is to fit the GPD pdf in equation 2 to the final loss distribution and use it to increase the emphasis placed on the tail samples during training. We denote the loss function of a given model, base loss, as $l$. In probabilistic forecasting, a commonly used loss is Negative Log Likelihood (NLL) loss: $l_i = -\log(p(\mathbf{y}^{(i)}|\mathbf{x}^{(i)}))$ where $\langle\mathbf{x}^{(i)}, \mathbf{y}^{(i)}\rangle$ is the $i^{th}$ training sample.

Our goal is to reduce the long-tail error measured by, e.g. MSE. This means that using NLL to fit the GPD might not lead to the intended prioritization of samples. Thus, we propose using an auxiliary loss $\hat{l}$, which is better correlated with the evaluation metric used, to fit the GPD. The choice of auxiliary loss is completely up to the model designer and could be the base loss itself in settings where it correlates well with the evaluation metric. See Appendix F for further details.

There are two main classes of loss augmentation methods to mitigate tail errors: *regularization* Ren et al. (2020); Makansi et al. (2021) and *reweighting* Lin et al. (2017); Lu et al. (2018); Yang et al. (2021). Inspired by these, we propose two variations of the Pareto Loss using the GPD fitted on $\hat{l}$: Pareto Loss Margin (PLM) and Pareto Loss Weighted (PLW).

PLM is based on the principles of margin-based regularization Ren et al. (2020); Liu et al. (2016), which assigns larger additive penalties to tail samples using the fitted GPD. For a given hyperparameter $\lambda$, PLM is defined as,

$$l_{plm} = l + \lambda * r_{plm}(\hat{l}), \quad r_{plm}(\hat{l}) = 1 - f_{(\xi,\eta)}(\hat{l}) \tag{3}$$

An alternative is to reweigh the loss terms using the fitted GPD. For a given $\lambda$, PLW is defined as,

$$l_{plw} = w_{plw}(\hat{l}) * l, \quad w_{plw}(\hat{l}) = 1 - \lambda * f_{(\xi,\eta)}(\hat{l}) \tag{4}$$

## 3.3 Kurtosis Loss

Use cases requiring higher emphasis on the extreme tail need an even more skewed measure of heavy-tailedness. For such cases we propose using Kurtosis, which is the scaled fourth moment relative to its mean. It assesses the propensity of a distribution to have extreme values within its tails. To increase the emphasis on tail samples, we use this measure as a margin-based regularization term in our proposed Kurtosis Loss. For a given hyperparameter $\lambda$ and using the same notations as Sec.3.2, Kurtosis Loss is defined as,

$$l_{kurt} = l + \lambda * r_{kurt}(\hat{l}), \quad r_{kurt}(\hat{l}) = \left(\frac{\hat{l} - \mu_{\hat{l}}}{\sigma_{\hat{l}}}\right)^4 \tag{5}$$

where $\mu_{\hat{l}}$ and $\sigma_{\hat{l}}$ are the mean and standard deviation of the auxiliary loss ($\hat{l}$) for a batch of samples. We do not use a reweighting based approach with kurtosis as there is no upper bound to the kurtosis value. This could lead to convergence issues due to very high weights for some samples.

## 3.4 Connection between Pareto and Kurtosis Loss

Kurtosis Loss and Pareto Loss are both based on moments of a distribution. Pareto Loss is a weighted sum of shifted moments, while Kurtosis Loss is the scaled fourth moment. Specifically, let $b = \frac{\xi a}{\eta}$ and $c = -(\frac{1}{\xi} + 1)$, then the Taylor expansion for the GPD pdf in equation 2 is,

$$(1+b)^c = 1 + cb + \frac{c(c-1)}{2!}b^2 + \frac{c(c-1)(c-2)}{3!}b^3 + \cdots \tag{6}$$

For $c < 0$ or equivalently $\xi < -1$ or $\xi > 0$, the coefficients are positive for even moments and negative for odd moments (odd and even powers of b). Even moments are always symmetric and positive, whereas odd moments are positive only for right-tailed distributions. Since we use the negative of the pdf, it yields an asymmetric measure of the right-tailedness of the distribution.

Kurtosis Loss uses the fourth moment. This is a symmetric and positive measure. GPD and kurtosis are visualized in Appendix E. Kurtosis emphasizes extreme values in the tail. Our experiments also show that it is more effective in controlling the extremes in the error distribution.

## 4 EXPERIMENTS

We evaluate our methods on multiple benchmark datasets from two probabilistic forecasting tasks: time series forecasting (1D) and trajectory prediction (2D).

### 4.1 SETUP

**Datasets.** For time series forecasting, we use electricity and traffic datasets from the UCI ML repository Dua & Graff (2017) used in Salinas et al. (2020) as benchmarks. We also generate three synthetic 1D time series datasets, Sine, Gaussian, and Pareto, to further our understanding of potential causes of long-tail error distribution. For trajectory prediction, we use two benchmark datasets: a pedestrian trajectory dataset ETH-UCY (which is a combination of ETH Pellegrini et al. (2009) and UCY Lerner et al. (2007) datasets) and a vehicle trajectory dataset nuScenes Caesar et al. (2020). Further details regarding the datasets are available in Appendix A.

**Baselines.** We compare with SoTA baselines in long tail mitigation for different tasks:

- Contrastive Loss: Makansi et al. (2021) uses contrastive loss as a regularizer to group examples together. The grouping is based on Kalman Filter prediction errors as a measure of sample difficulty.
- Label Distribution Smoothing (LDS): Yang et al. (2021) uses a symmetric kernel to smooth the label distribution and use its inverse to reweigh the loss terms.
- Shrinkage Loss: Lu et al. (2018) uses a sigmoid-based function to reweigh loss terms. This deprioritizes lower loss values.
- Focal Loss: Lin et al. (2017) uses L1 loss to reweigh the loss terms. Additional power of the loss term increases the steepness of the loss function.

Focal Loss, Shrinkage Loss, and LDS were originally proposed for classification and/or regression and required adaptation to be applied to the forecasting task. See Appendix B for details.

**Evaluation Metrics.** We use metrics in accordance with literature Walters et al. (2021); Salzmann et al. (2020); Makansi et al. (2021): Average Displacement Error (ADE), which is the average L2 distance between total predicted trajectory and ground truth, and Final Displacement Error (FDE) which is the L2 distance for the final timestep. For time series forecasting, we use the metrics from DeepAR Salinas et al. (2020) and use Normalized Deviation (ND) and Normalized Root Mean Squared Error (NRMSE). We also report Continuous Ranked Probability Score (CRPS) Gneiting & Ranjan (2011) for the time series datasets, a more suitable metric for probabilistic forecasting.

Apart from the above-mentioned average performance metrics, we introduce metrics to capture the tail errors. We adapt the Value-at-Risk (VaR equation 7) tail metric from financial domain:

$$\text{VaR}_\alpha(E) = \inf\{e \in E : P(E \geq e) \leq 1 - \alpha\} \tag{7}$$

VaR at level $\alpha \in (0, 1)$ is the smallest error $e$ such that the probability of observing error greater than $e$ is less than $1 - \alpha$, where $E$ is the error distribution. This evaluates to the $\alpha^{th}$ quantile of the error distribution. We measure VaR at three different levels: 0.95, 0.98, and 0.99. Additionally, we report the maximum error representing the worst-case performance. We present tail metrics on the complete error distribution as there is no fixed set of tail samples across different methods (See Sec.3.1).

### 4.2 SYNTHETIC DATASET EXPERIMENTS

To better understand the long tail in error, we perform experiments on three synthetic datasets. The task is to forecast 8 steps ahead given a history of 8 time steps. We use AutoRegression (AR) and DeepAR Salinas et al. (2020) as models to perform this task. The top row in Figure 3 shows that

among the datasets, only Gaussian and Pareto exhibit tail in the data distribution. The data distribution is available here only because the datasets were generated synthetically.

On the Sine dataset, we observe long tail error for DeepAR but not for AR. This is especially significant as there is *no long tail in the data distribution*. On Gaussian and Pareto datasets, DeepAR leads to a heavier tail than AR, suggesting that the long tail in data also contributes to long tail in error. The difference between AR and DeepAR error distributions also invalidates the assumption made by Makansi et al. (2021)). Using the prediction performance from Kalman Filter is not a good indicator of sample tailedness for deep neural networks. The complete results for synthetic datasets are available in appendix K.

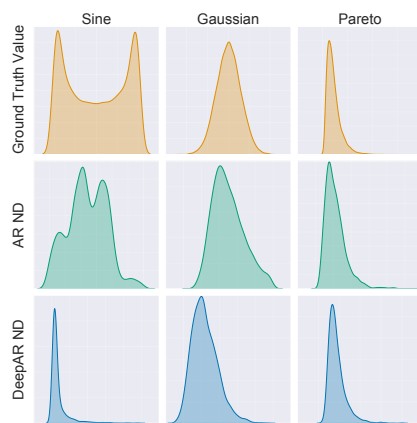

Figure 3: Top Row: Ground truth distribution for synthetic datasets. Middle Row: ND error distribution using AR. Bottom Row : ND error distribution using DeepAR. Datasets (L to R): Sine, Gaussian, Pareto. **Note**: the x-axes for plots in the same column or y-axes for plots in the same row are not for the same range of values.

### 4.3 REAL-WORLD EXPERIMENTS

**Time Series Forecasting.** We present average and tail metrics using ND and NRMSE for the time series forecasting task on electricity and traffic datasets in Tables 1 and 3 respectively. All methods use DeepAR Salinas et al. (2020), one of the SoTA in probabilistic time series forecasting, as the base model. The task for both datasets is to use a 1-week history (168 hours) to forecast for 1 day (24 hours) at an hourly frequency. The base model exhibits long tail behavior in error on both datasets (see Fig. 2). The tail of the error distribution is significantly longer for the traffic dataset as compared to the electricity dataset. This is evident from comparing the tail error values to the average error. The auxiliary loss used here is MAE to correlate with L1 metrics like ND. DeepAR can have intrinsic variation on re-training so results in Table 1 are averaged over 3 runs.

**Trajectory Forecasting.** We present experimental results on ETH-UCY and nuScenes datasets in Tables 2 and 4 respectively. Following Salzmann et al. (2020) and Makansi et al. (2021) we calculate model performance based on the best out of 20 guesses. On both datasets, we compare with several long-tail baselines using Trajectron++EWTA Makansi et al. (2021) as a base model due to its SoTA average performance on these datasets. The auxiliary loss used here is MAE with MSE to correlate with L2 metrics like ADE and FDE.

### 4.4 RESULTS ANALYSIS

**Cross-task consistency.** As shown in Tables 1, 3, 2 and 4, our proposed approaches, Kurtosis Loss and PLM, are the only methods improving on tail metrics across all tasks. Our methods typically deliver 10-15% improvement on tail metrics and sometimes as high as 40% (See Appendix G). These are significant improvements with no sacrifice on average performance for any task. In fact, in some tasks our methods have better average performance as well.

The generality of our methods is shown by their success on all studied tasks. Our tasks have different base models (DeepAR, Trajectron++EWTA), data representations (1D: Time series, 2D: Trajectory), base losses (GaussianNLL for Time series, EWTA for Trajectory), and forecasting horizons. Our methods provide consistent improvement on tail metrics for all tasks. In comparison, Focal Loss performs well on trajectory datasets but fails on time series datasets. Contrastive Loss only performs well on Traffic dataset. LDS and Shrinkage Loss do not compare to the best results for any dataset and perform worse than the base model on the time series datasets.

We illustrate some difficult examples, examples with large errors common across methods, for all real world datasets in Figure 4 to demonstrate the improvement in the quality of forecast for our methods.

**Re-weighting vs Regularization.** As mentioned in Section 3.2, we can categorize loss modifying methods into two classes: re-weighting (Focal Loss, Shrinkage Loss, LDS and PLW) and regulariza-

Table 1: Performance on **Electricity Dataset** (ND/NRMSE/CRPS). All our methods improve on the average as well as tail metrics. Baseline methods are worse on average and inconsistent on the tail. All methods use DeepAR as the base model. Results indicated as Top 3 and **Best**. All results have been averaged across 3 runs with different seeds, standard deviation available in Appendix H

| METHOD | METRIC | MEAN↓ | VaR$_{95}$ ↓ | VaR$_{98}$ ↓ | VaR$_{99}$ ↓ | MAX↓ |
|---|---|---|---|---|---|---|
| BASE MODEL | ND | 0.0600 | 0.0793 | 0.2251 | 0.4356 | 4.2777 |
| | NRMSE | 0.3069 | **0.0991** | 0.2533 | 0.5430 | 5.5994 |
| | CRPS | 142 | 463 | 1138 | 1996 | 30705 |
| CONTRASTIVE LOSS | ND. | 0.0696 | 0.0954 | 0.2419 | 0.4646 | 4.5286 |
| | NRMSE | 0.3345 | 0.1138 | 0.2778 | 0.5504 | 5.6761 |
| | CRPS | 167 | 521 | 1266 | 2363 | 31835 |
| FOCAL LOSS | ND | 0.0639 | 0.0859 | 0.2505 | 0.4456 | 4.3217 |
| | NRMSE | 0.3110 | 0.1062 | 0.2922 | 0.5342 | 5.4843 |
| | CRPS | 150 | 474 | 1195 | 2103 | 30224 |
| SHRINKAGE LOSS | ND | 0.0673 | 0.0888 | 0.2328 | 0.4568 | 4.5911 |
| | NRMSE | 0.3247 | 0.1103 | 0.2871 | 0.5213 | 5.6334 |
| | CRPS | 156 | 480 | 1199 | 2240 | 28398 |
| LDS | ND | 0.0632 | 0.0920 | 0.2287 | 0.4620 | 3.8626 |
| | NRMSE | 0.2980 | 0.1152 | 0.2790 | 0.5322 | 5.0126 |
| | CRPS | 151 | 496 | 1185 | 2110 | 29959 |
| KURTOSIS LOSS (OURS) | ND | **0.0578** | 0.0827 | 0.2132 | 0.4044 | 3.6565 |
| | NRMSE | 0.2801 | 0.1023 | 0.2564 | 0.4958 | 4.7673 |
| | CRPS | **140** | 455 | 1105 | **1952** | 26946 |
| PLM (OURS) | ND | 0.0580 | **0.0791** | **0.2018** | 0.3990 | 3.7827 |
| | NRMSE | 0.2897 | 0.1011 | **0.2396** | **0.4844** | 5.0230 |
| | CRPS | 141 | **449** | 1111 | 2044 | 28992 |
| PLW (OURS) | ND | 0.0581 | 0.0793 | 0.2191 | **0.3917** | **3.5673** |
| | NRMSE | **0.2789** | 0.1013 | 0.2569 | **0.4973** | **4.7328** |
| | CRPS | **140** | 454 | **1099** | 1953 | **26273** |

Table 2: Macro-averaged performance on the **ETH-UCY Dataset** (ADE/FDE). Our approaches improve tail performance better than existing baselines. The improvements are most significant for far-future prediction (FDE). PLM improves well across prediction horizon (ADE). All methods utilize Trajectron++EWTA as the base model. Results indicated as Top 3 and **Best**.

| METHOD | MEAN↓ | VaR$_{95}$ ↓ | VaR$_{98}$ ↓ | VaR$_{99}$ ↓ | MAX↓ |
|---|---|---|---|---|---|
| BASE MODEL | **0.16**/0.33 | 0.43/1.05 | 0.60/1.53 | 0.76/1.89 | 1.63/3.95 |
| CONTRASTIVE | 0.17/0.34 | 0.43/1.03 | 0.62/1.56 | 0.79/1.89 | 1.67/4.02 |
| FOCAL LOSS | **0.16**/0.32 | 0.40/0.89 | 0.54/1.28 | 0.66/1.57 | 1.50/3.50 |
| SHRINKAGE LOSS | **0.16**/0.33 | 0.43/1.05 | 0.58/1.50 | 0.74/1.84 | 1.66/3.95 |
| LDS | 0.17/0.35 | 0.44/1.04 | 0.57/1.45 | 0.78/1.86 | 1.69/3.85 |
| KURTOSIS LOSS (OURS) | 0.17/0.34 | 0.46/0.98 | 0.59/1.25 | 0.67/1.47 | **1.22/2.77** |
| PLM (OURS) | **0.16/0.30** | **0.38/0.81** | **0.52**/1.20 | **0.63**/1.49 | 1.30/3.20 |
| PLW (OURS) | 0.21/0.36 | 0.46/0.84 | 0.55/**1.08** | **0.63/1.32** | 1.25/2.93 |

tion (Contrastive Loss, PLM and Kurtosis Loss). Re-weighting multiplies the loss for tail samples with higher weights. Regularization adds higher regularization values for samples with higher loss.

We notice that re-weighting methods perform worse as the long-tail in error worsens. In scenarios with longer tails, the weights of tail samples can be very high. Overemphasizing tail examples might hamper the learning for other samples. Notice the significantly worse average performance of Focal loss for the traffic dataset in Table 3. Shrinkage Loss limits this issue by bounding the weights but fails to show tail improvements in longer tail scenarios (electricity and traffic datasets). Our proposed PLW is the best reweighting method on most datasets, likely due to bounded weights.

Table 3: Performance on the **Traffic Dataset** (ND/NRMSE/CRPS). PLM (Ours) delivers best overall results, improving on average and tail metrics. Among baseline methods, contrastive loss is most consistent. Regularization methods in general fare better than re-weighting methods due to a very long tail. All methods use DeepAR as the base model. Results indicated as Top 3 and **Best**

| METHOD | METRIC | MEAN↓ | VaR$_{95}$ ↓ | VaR$_{98}$ ↓ | VaR$_{99}$ ↓ | MAX↓ |
|---|---|---|---|---|---|---|
| BASE MODEL | ND | 0.1741 | **0.6866** | 25.5840 | 32.1330 | 84.1582 |
| | NRMSE | **0.4465** | **1.2283** | 6.0283 | 7.5988 | 18.8103 |
| | CRPS | 0.0068 | 0.0211 | 0.0412 | 0.0691 | 0.8524 |
| CONTRASTIVE LOSS | ND | 0.2052 | 0.7463 | **24.3737** | 30.5117 | 81.1716 |
| | NRMSE | 0.4667 | 1.2956 | 5.7747 | 7.2342 | 18.3360 |
| | CRPS | 0.0079 | 0.0235 | 0.0450 | 0.0802 | 0.8517 |
| FOCAL LOSS | ND | 0.4903 | 1.1553 | 26.7537 | **30.1506** | **52.8272** |
| | NRMSE | 0.7302 | 1.6485 | 6.5880 | 7.3660 | 13.7985 |
| | CRPS | 0.0183 | 0.0463 | 0.0639 | 0.0933 | 0.8471 |
| SHRINKAGE LOSS | ND | 0.2431 | 0.8380 | 25.3381 | 32.9147 | 85.2713 |
| | NRMSE | 0.5114 | 1.3099 | 6.0418 | 7.8882 | 19.0771 |
| | CRPS | 0.0093 | 0.0316 | 0.0511 | 0.0732 | 0.8573 |
| LDS | ND | 0.4763 | 1.4781 | 28.9162 | 38.4263 | 126.5733 |
| | NRMSE | 0.7829 | 1.8702 | 6.8826 | 9.2061 | 27.3684 |
| | CRPS | 0.0175 | 0.0564 | 0.0802 | 0.1074 | 0.8530 |
| KURTOSIS LOSS (OURS) | ND | 0.2022 | 0.7653 | 25.3752 | 31.4677 | 62.9173 |
| | NRMSE | 0.4892 | 1.4072 | 6.0263 | 7.3369 | **13.7783** |
| | CRPS | 0.0081 | 0.0243 | **0.0409** | **0.0682** | 0.8491 |
| PLM (OURS) | ND | **0.1594** | 0.7115 | 24.5911 | 30.331 | 90.3169 |
| | NRMSE | 0.4600 | 1.3881 | **5.6779** | **7.0033** | 20.5736 |
| | CRPS | **0.0065** | **0.0185** | 0.0429 | 0.0822 | **0.8463** |
| PLW (OURS) | ND | 0.3751 | 1.0495 | 25.4471 | 31.6621 | 65.759 |
| | NRMSE | 0.6238 | 1.4914 | 6.0552 | 7.3491 | 13.8938 |
| | CRPS | 0.0126 | 0.0361 | 0.0501 | 0.0716 | 0.8571 |

Table 4: Average performance on the **nuScenes Dataset** (ADE/FDE). Our approaches improve tail performance for far-future prediction (FDE) better than existing baselines. All methods utilize Trajectron++EWTA as the base model. Results indicated as Top 3 and **Best**.

| METHOD | MEAN↓ | VaR$_{95}$ ↓ | VaR$_{98}$ ↓ | VaR$_{99}$ ↓ | MAX↓ |
|---|---|---|---|---|---|
| BASE MODEL | **0.19**/0.34 | 0.65/1.49 | 1.00/2.49 | 1.32/3.34 | 7.07/11.42 |
| CONTRASTIVE | **0.19**/0.35 | 0.65/1.51 | 1.01/2.58 | 1.36/3.46 | 6.82/10.48 |
| FOCAL LOSS | **0.19**/0.33 | **0.56**/1.09 | 0.85/1.95 | 1.11/2.65 | 6.55/11.71 |
| SHRINKAGE LOSS | **0.19**/**0.32** | 0.62/1.32 | 0.96/2.31 | 1.25/3.17 | 6.39/10.26 |
| LDS | **0.19**/**0.32** | 0.62/1.26 | 0.94/2.23 | 1.20/2.99 | **5.20**/10.53 |
| KURTOSIS LOSS (OURS) | 0.20/0.38 | 0.65/1.35 | 0.85/1.82 | 1.03/2.27 | 5.39/**7.52** |
| PLM (OURS) | **0.19**/0.33 | 0.62/1.32 | 0.95/2.31 | 1.25/3.18 | 6.10/10.96 |
| PLW (OURS) | 0.24/0.37 | 0.60/**1.00** | **0.82**/**1.49** | **1.01**/**2.01** | 7.51/9.91 |

In contrast, regularization methods are consistent across all tasks on both tail and average metrics. The additive nature of regularization limits the impact tail samples have on the learning. This enables these methods to handle different severities of long-tail without degrading the average performance.

**Choosing between PLM and Kurtosis Loss** Kurtosis Loss performs better on extreme tail metrics, VaR$_{99}$ and Max. Higher kurtosis puts more emphasis on extreme samples in the tail. It is also important to note that the magnitude of kurtosis varies significantly for different distributions, making the choice of hyperparameter (See equation 5) critical. Further analysis available in Appendix D.

PLM is the most consistent method across all tasks. As noted by McNeil (1997) GPD is well suited to model long tail error distributions. PLM rewards examples moving away from the tail towards the mean with significantly lower margin values. PLM margin values saturate beyond a point in the tail providing similar penalties for long-tail samples. Comparatively, Kurtosis Loss is sensitive to

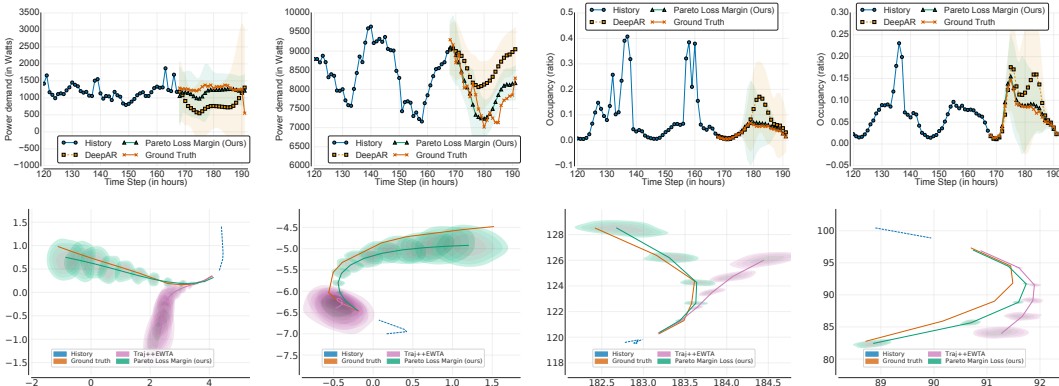

Figure 4: Visualization of overlapping tail samples for Electricity (top row left half), Traffic (top row right half), ETH-UCY (bottom row left half) and nuScenes (bottom row right half) datasets. The shaded region represents the confidence interval of the prediction. The difficulty here is a departure from historical behavior. This manifests as sudden increases or decreases in the 1D time series datasets and as high velocity trajectories with sharp turns for the trajectory datasets. These samples represent critical events in real world scenarios where the performance of the model is of utmost importance. Our methods perform significantly better on such samples.

extreme samples in the tail. This shows in performance with Kurtosis Loss performing better on $\text{VaR}_{99}$ and Max, and PLM performing better on $\text{VaR}_{95}$ and $\text{VaR}_{98}$. The choice between the methods depends on the objective. If the preference is to mitigate extreme samples, then Kurtosis Loss is better. Otherwise, if the preference is to improve on the tail overall, then PLM is better.

**Tail error and long-term forecasting.** Based on the trajectory prediction results in Tables 2 and 4 we observe that the error reduction for tail samples is more visible in FDE than in ADE. This indicates that the magnitude of the observed error increases with the forecasting horizon. The error compounds through prediction steps making far-future predictions inherently more difficult. Larger improvements in FDE indicate that both Kurtosis and Pareto Loss ensure that high tail errors (stemming from large, far-future prediction errors measured by FDE) are decreased.

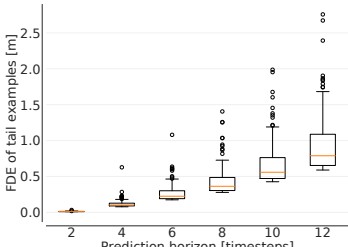

Figure 5: Distribution of the top 5% error values (FDE) for different horizons for the ETH-UCY (Zara1) dataset. Predictions obtained using Trajectron++EWTA. The trend shows that the long tail in error gets worse as the forecasting horizon increases due to compounding.

Accurate long-term forecasting is a central goal of deep probabilistic forecasting. As we can see in Fig. 5, the tail of error distribution is more pronounced with longer horizons. Thus, methods addressing the tail performance are necessary in order to ensure the practical applicability and reliability of future long-term prediction models.

## 5 CONCLUSION

We identify and address the problem of long-tail in error distribution for deep probabilistic forecasting. We propose Pareto Loss (Margin and Weighted) and Kurtosis Loss, two novel moment-based loss augmentation approaches, increasing emphasis on tail samples adaptively. We demonstrate their practical effects on two spatiotemporal trajectory datasets and two time series datasets using different base models. Our methods achieve significant improvements on tail metrics over existing baselines without degrading average performance. Both proposed losses can be easily integrated with existing approaches in deep probabilistic forecasting to improve their performance on tail metrics.

Future directions include more principled ways to tune hyperparameters, extensions to deterministic time series forecasting models, and theoretical analysis for the source of the long-tail error. Based on our observations, we suggest evaluating tail metrics apart from average performance in machine learning tasks to identify potential long tail issues across different tasks and domains.

## REPRODUCIBILITY STATEMENT

The datasets used in the paper are cited and the preprocessing has been described in Appendix A. We have released the code to run experiments on both time series and trajectory datasets in the supplementary material. Both folders include a step-by-step README file that guides through the process of running our methods and baselines. Hyperparamter values to be used are present in Appendix D. We have also provided the location of the base code that was used in Appendix C.

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

## A    DATASET DESCRIPTION

The ETH-UCY dataset consists of five subdatasets, each with Bird's-Eye-Views: ETH, Hotel, Univ, Zara1, and Zara2. As is common in the literature Makansi et al. (2021); Salzmann et al. (2020) we present macro-averaged 5-fold cross-validation results in our experiment section. The nuScenes dataset includes 1000 scenes of 20 second length for vehicle trajectories recorded in Boston and Singapore.

The electricity dataset contains electricity consumption data for 370 homes over the period of Jan 1st, 2011 to Dec 31st, 2014 at a sampling interval of 15 minutes. We use the data from Jan 1st, 2011 to Aug 31st, 2011 for training and data from Sep 1st, 2011 to Sep 7th, 2011 for testing. The traffic dataset consists of occupancy values recorded by 963 sensors at a sampling interval of 10 minutes ranging from Jan 1st, 2008 to Mar 30th, 2009. We use data from Jan 1st, 2008 to Jun 15th, 2008 for training and data from Jun 16th, 2008 to Jul 15th, 2008 for testing. Both time series datasets are downsampled to 1 hour for generating examples.

The synthetic datasets are generated as 100 different time series consisting of 960 time steps. Each time series in the Sine dataset is generated using a random offset $\theta$ and a random frequency $\nu$ both selected from a uniform distribution $U[0, 1]$. Then the time series is $sin(2\pi\nu t + \theta)$ where $t$ is the index of the time step. Gaussian and Pareto datasets are generated as order 1 lag autoregressive time series with randomly sampled Gaussian and Pareto noise respectively. Gaussian noise is sampled from a Gaussian distribution with mean 1 and standard deviation 1. Pareto noise is randomly sampled from a Pareto distribution with shape 10 and scaling 1.

## B    METHOD ADAPTATION

**Time Series forecasting**    DeepAR uses Gaussian Negative Log Likelihood as the loss which is unbounded. Due to this many baseline methods need to be adapted in order to be usable. For the same reason, we also need an auxiliary loss $(\hat{l})$. We use MAE loss to fit the GPD, calculate kurtosis, and to calculate the weight terms for Focal and Shrinkage loss. For LDS we treat all labels across time steps as a part of a single distribution. Additionally, to avoid extremely high weights $(\mathcal{O}(10^8))$ in LDS due to the nature of long tail we ensure a minimum probability of $0.001$ for all labels.

**Trajectory forecasting**    We adapt Focal Loss and Shrinkage Loss to use EWTA loss Makansi et al. (2019) in order to be compatible with Trajectron++EWTA base model. LDS was originally proposed for a regression task and we adapt it to the trajectory prediction task in the same way as for the time series task. We use MAE to fit the GPD, due to the Evolving property of EWTA loss.

## C    IMPLEMENTATION DETAILS

**Time    Series    forecasting**  We    use    the    DeepAR    implementation    from https://github.com/zhykoties/TimeSeries as the base code to run all time series experiments. The original code is an AWS API and not publicly available. The implementation of contrastive loss is taken directly from the source code of Makansi et al. (2021).

**Trajectory forecasting**    For the base model of Trajectron++EWTA Makansi et al. (2021) we have used the original implementation provided by the original authors. The implementation of contrastive loss is taken directly from the source code of Makansi et al. (2021).

The experiments have been conducted on a machine with 7 RTX 2080 Ti GPUs.

## D    HYPERPARAMETER TUNING

We observe during our experiments that the performance of Kurtosis Loss is highly dependent on the hyperparameter $\lambda$ (See equation 5). Results for different values of $\lambda$ on the electricity dataset for Kurtosis Loss are shown in Table5. We also show the variation of ND and NRMSE with the hyperparameter value in Figure 6. We can see that there is an optimal value of the hyperparameter and the approach performs worse with higher and lower values.

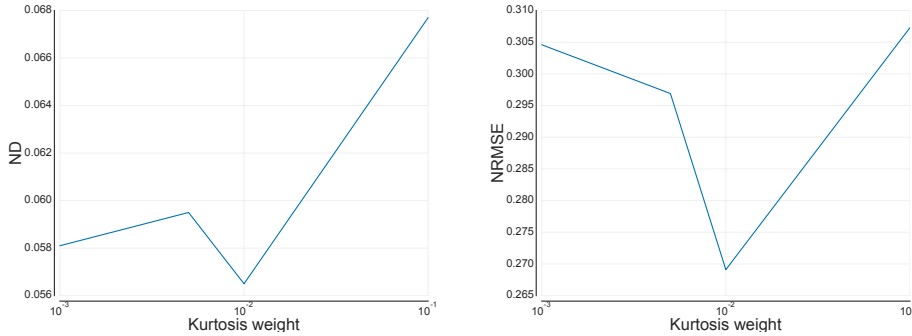

Figure 6: Left: Variation of ND by hyperparameter for Kurtosis Loss. Right: Variation of NRMSE by hyperparameter for Kurtosis Loss.

For both ETH-UCY and nuScenes datasets we have used $\lambda = 0.1$ for Kurtosis Loss, and $\lambda = 1$ for PLM and PLW. For both electricity and traffic datasets, we use $\lambda = 1$ for PLM, $\lambda = 0.5$ for PLW and $\lambda = 0.01$ for Kurtosis Loss.

Table 5: Electricity Dataset evaluation for base model (ND/NRMSE) and different Kurtosis Loss hyperparameters. The value of $\lambda$ is denoted in [] with the method name. The base model is DeepAR. Results indicated as Better than base model and **Best**

| METHOD | METRIC | MEAN$\downarrow$ | $VAR_{95}\downarrow$ | $VAR_{98}\downarrow$ | $VAR_{99}\downarrow$ | MAX$\downarrow$ |
|---|---|---|---|---|---|---|
| BASE MODEL | ND | 0.0584 | 0.0796 | 0.2312 | 0.4429 | 4.1520 |
| | NRMSE | 0.2953 | **0.0972** | 0.2595 | 0.5263 | 5.4950 |
| KURTOSIS LOSS [0.001] | ND | 0.0581 | 0.0815 | **0.2087** | **0.3936** | 4.2381 |
| | NRMSE | 0.3046 | 0.1014 | **0.2325** | **0.4756** | 5.7144 |
| KURTOSIS LOSS [0.005] | ND | 0.0574 | **0.0767** | 0.2147 | 0.4138 | 3.6767 |
| | NRMSE | 0.2843 | 0.0999 | 0.2617 | 0.4792 | 5.0062 |
| KURTOSIS LOSS [0.01] | ND | **0.0567** | 0.0842 | 0.2151 | 0.4120 | **3.2738** |
| | NRMSE | **0.2631** | 0.1046 | 0.2732 | 0.4779 | **4.2613** |
| KURTOSIS LOSS [0.1] | ND | 0.0677 | 0.0954 | 0.2269 | 0.4579 | 3.8772 |
| | NRMSE | 0.3073 | 0.1184 | 0.2768 | 0.5419 | 5.1345 |

## E PARETO AND KURTOSIS

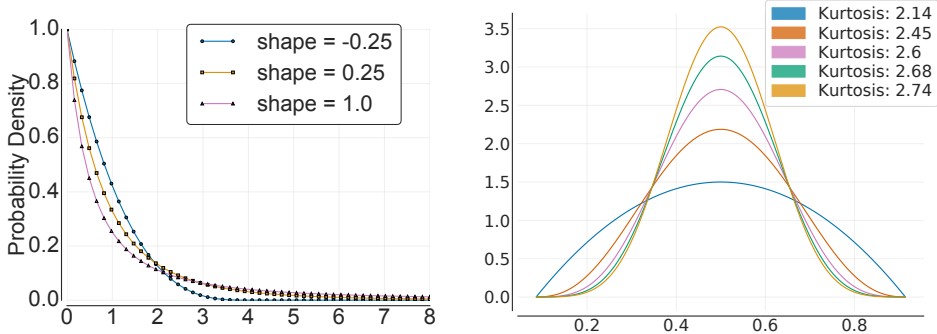

Figure 7: Left: Generalized Pareto distributions with different shape parameters ($\eta = 1$). Right: Illustrating the variation of kurtosis on distributions with the same mean.

Figure 7 illustrates different GPDs for different shape parameter values. Higher shape value models more severe tail behavior.

# F AUXILIARY LOSS

In this section, we present mathematical intuition behind the usage of auxiliary loss in our methods. We will examine a setting where the base loss for a probabilistic model is GaussianNLL loss and the evaluation metric is MSE. For simplicity, we will assume 1-step prediction on 1D data however the analysis can be easily extended to multi step prediction and multi dimensional data.

Consider 2 training samples,

Past observations : $\mathbf{x}^{(1)} = (x_{t-k}^{(1)}, \ldots, x_t^{(1)})$; $\mathbf{x}^{(2)} = (x_{t-k}^{(2)}, \ldots, x_t^{(2)})$

1-step prediction ground truth : $\mathbf{y}^{(1)} = (y_{t+1}^{(1)})$; $\mathbf{y}^{(2)} = (y_{t+1}^{(2)})$

Model prediction : $\mu_{t+1}^{(1)}, \sigma_{t+1}^{(1)}; \mu_{t+1}^{(2)}, \sigma_{t+1}^{(2)}$

We will drop t+1 from the notation for simplicity and clarity as there is only one step prediction. Since, the maximum likelihood prediction for a gaussian is the mean, the MSE is calculated using the predicted mean.

$$\text{MSE} : (y^{(i)} - \mu^{(i)})^2 \tag{8}$$

The GaussianNLL loss is calculated as the negative log likelihood of the ground truth as per the predicted distribution. Simplifying the expression gives us,

$$\text{GaussianNLL loss} : \ln\left(\sigma^{(i)}\sqrt{2\pi}\right) + \frac{1}{2}\left(\frac{y^{(i)} - \mu^{(i)}}{\sigma^{(i)}}\right)^2 \tag{9}$$

We want to determine the conditions under which the GaussianNLL loss will be higher for sample 1 as compared to sample 2 while the MSE for sample 2 will be higher than sample 1 or vice versa. We will call this a loss-metric inversion. This condition can be written as:

$$(GaussianNLL^{(1)} - GaussianNLL^{(2)})(MSE^{(1)} - MSE^{(2)}) < 0 \tag{10}$$

Consider the scenario where, $MSE^{(1)} > MSE^{(2)}$. This can be expressed as,

$$(y^{(1)} - \mu^{(1)})^2 = k(y^{(2)} - \mu^{(2)})^2 \text{ where } k > 1 \qquad \text{(From equation 8)} \tag{11}$$

The corresponding condition to satisfy is,

$$(GaussianNLL^{(1)} - GaussianNLL^{(2)}) < 0 \qquad \text{(From equation 10)}$$

$$\implies \ln(\frac{\sigma^{(1)}}{\sigma^{(2)}}) + \frac{1}{2}\left(\left(\frac{y^{(1)} - \mu^{(1)}}{\sigma^{(1)}}\right)^2 - \left(\frac{y^{(2)} - \mu^{(1)}}{\sigma^{(2)}}\right)^2\right) < 0 \qquad \text{(From equation 9)}$$

$$\implies \frac{1}{2}(y^{(2)} - \mu^{(2)})^2 \left(\frac{k}{\sigma^{(1)2}} - \frac{1}{\sigma^{(2)2}}\right) < \ln(\frac{\sigma^{(2)}}{\sigma^{(1)}}) \qquad \text{(From equation 11)}$$

Consider, $\sigma^{(1)} = c\sigma^{(2)}$, where $c > 0$

$$\frac{1}{2}\left(\frac{y^{(2)} - \mu^{(2)}}{\sigma^{(2)}}\right)^2 \left(\frac{k}{c^2} - 1\right) < \ln(\frac{1}{c})$$

For simplicity let's represent $\frac{1}{2}\left(\frac{y^{(2)} - \mu^{(2)}}{\sigma^{(2)}}\right)^2$ as a single variable $m$.

$$m\left(\frac{k}{c^2} - 1\right) + \ln(c) < 0$$

For a fixed $k$ the minima for the LHS is achieved for $c = \sqrt{2km}$. The value of the LHS at minima is,

$$\left(\frac{1}{2} - m\right) + \frac{1}{2}\ln(km) = \frac{1}{2}\ln\left(\frac{km}{e^{2m-1}}\right)$$

Since the numerator in the log form is linear in $m$ and the denominator is exponential in $m$ the minima can be less than zero for suitable values of $m$.

This shows that there can be pairs of samples with loss-metric inversion. This means that regularization and reweighting values can be completely different from intended unless an auxiliary loss is used, which preserves the order w.r.t. the evaluation metric. This lack of correlation is illustrated in Fig 8 for the DeepAR model on the electricity dataset.

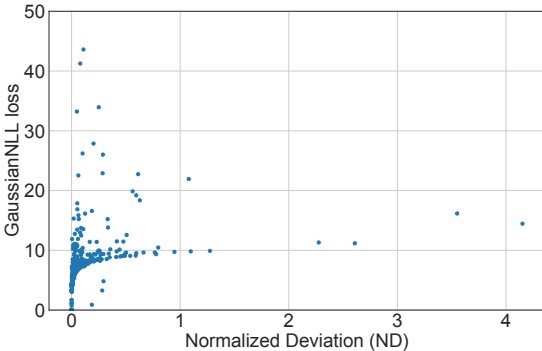

Figure 8: Comparing GaussianNLL loss to Normalized Deviation metric for DeepAR on the electricity dataset. We can see that there are a large number of samples which have high GaussianNLL but low ND and vice versa. This illustrates the need of an auxiliary loss for correct emphasis on samples.

## G  PERCENTAGE IMPROVEMENTS

We present percentage improvements compared to the base model for the different datasets.

Table 6: Percentage improvements over the base method (DeepAR) on **Electricity Dataset** (ND/NRMSE). Results indicated as error reduction and increase in %.

| METHOD | METRIC | MEAN↓ | VaR$_{95}$ ↓ | VaR$_{98}$ ↓ | VaR$_{99}$ ↓ | MAX↓ |
|---|---|---|---|---|---|---|
| CONTRASTIVE LOSS | ND | 15.94 | 20.26 | 7.43 | 6.67 | 5.86 |
| | NRMSE | 8.99 | 14.79 | 9.66 | 1.24 | 1.37 |
| FOCAL LOSS | ND | 6.44 | 8.32 | 11.27 | 2.30 | 1.03 |
| | NRMSE | 1.34 | 7.16 | 15.34 | 1.72 | 2.06 |
| SHRINKAGE LOSS | ND | 12.17 | 11.94 | 3.42 | 4.86 | 7.33 |
| | NRMSE | 5.80 | 11.26 | 13.33 | 4.10 | 0.61 |
| LDS | ND | 5.28 | 16.06 | 1.57 | 6.06 | 9.71 |
| | NRMSE | 2.90 | 16.21 | 10.13 | 2.10 | 10.48 |
| KURTOSIS LOSS (OURS) | ND | 3.72 | 4.33 | 5.29 | 7.16 | 14.52 |
| | NRMSE | 8.72 | 3.16 | 1.21 | 8.80 | 14.86 |
| PLM (OURS) | ND | 3.28 | 0.21 | 10.36 | 8.40 | 11.57 |
| | NRMSE | 5.58 | 1.98 | 5.41 | 10.89 | 10.29 |
| PLW (OURS) | ND | 3.22 | 0.00 | 2.68 | 10.09 | 16.61 |
| | NRMSE | 9.10 | 2.19 | 1.42 | 8.52 | 15.48 |

Table 7: Percentage improvements over the base method (DeepAR) on **Traffic Dataset** (ND/NRMSE). Results indicated as error reduction and increase in %.

| METHOD | METRIC | MEAN↓ | VaR$_{95}$ ↓ | VaR$_{98}$ ↓ | VaR$_{99}$ ↓ | MAX↓ |
|---|---|---|---|---|---|---|
| CONTRASTIVE LOSS | ND | 17.86 | 8.70 | 4.73 | 5.05 | 3.55 |
| | NRMSE | 4.52 | 5.48 | 4.21 | 4.80 | 2.52 |
| FOCAL LOSS | ND | 181.62 | 68.26 | 4.57 | 6.17 | 37.23 |
| | NRMSE | 63.54 | 34.21 | 9.28 | 3.06 | 26.64 |
| SHRINKAGE LOSS | ND | 39.63 | 22.05 | 0.96 | 2.43 | 1.32 |
| | NRMSE | 14.54 | 6.64 | 0.22 | 3.81 | 1.42 |
| LDS | ND | 173.58 | 115.28 | 13.02 | 19.59 | 50.40 |
| | NRMSE | 75.34 | 52.26 | 14.17 | 21.15 | 45.50 |
| KURTOSIS LOSS (OURS) | ND | 16.14 | 11.46 | 0.82 | 2.07 | 25.24 |
| | NRMSE | 9.56 | 14.56 | 0.03 | 3.45 | 26.75 |
| PLM (OURS) | ND | 8.44 | 3.63 | 3.88 | 5.61 | 7.32 |
| | NRMSE | 3.02 | 13.01 | 5.81 | 7.84 | 9.37 |
| PLW (OURS) | ND | 115.45 | 52.85 | 0.54 | 1.47 | 21.86 |
| | NRMSE | 39.71 | 21.42 | 0.45 | 3.29 | 26.14 |

Table 8: Percentage improvements over the base method (Trajectron++EWTA) on **ETH-UCY Dataset** (ADE/FDE). Results indicated as error reduction and increase in %.

| METHOD | MEAN↓ | VaR$_{95}$ ↓ | VaR$_{98}$ ↓ | VaR$_{99}$ ↓ | MAX↓ |
|---|---|---|---|---|---|
| CONTRASTIVE | 6.25/3.03 | 0.00/1.90 | 3.33/1.96 | 3.95/0.00 | 2.45/1.77 |
| FOCAL LOSS | 0.00/3.03 | 6.98/15.24 | 10.00/16.34 | 13.16/16.93 | 7.98/11.39 |
| SHRINKAGE LOSS | 0.00/0.00 | 0.00/0.00 | 3.33/1.96 | 2.63/2.65 | 1.84/0.00 |
| LDS | 6.25/6.06 | 2.33/0.95 | 5.00/5.23 | 2.63/1.59 | 3.68/2.53 |
| KURTOSIS LOSS (OURS) | 6.25/3.03 | 6.98/6.67 | 1.67/18.30 | 11.84/22.22 | 25.15/29.87 |
| PLM (OURS) | 0.00/9.09 | 11.63/22.86 | 13.33/21.57 | 17.11/21.16 | 20.25/18.99 |
| PLW (OURS) | 31.25/9.09 | 6.98/20.00 | 8.33/29.41 | 17.11/30.16 | 23.31/25.82 |

Table 9: Percentage improvements over the base method (Trajectron++EWTA) on **nuScenes Dataset** (ADE/FDE). Results indicated as error reduction and increase in %.

| METHOD | MEAN↓ | VaR$_{95}$ ↓ | VaR$_{98}$ ↓ | VaR$_{99}$ ↓ | MAX↓ |
|---|---|---|---|---|---|
| CONTRASTIVE | 0.00/2.94 | 0.00/1.34 | 1.00/3.61 | 3.03/3.59 | 3.54/8.23 |
| FOCAL LOSS | 0.00/2.94 | 13.85/26.85 | 15.00/21.69 | 15.91/20.66 | 7.36/2.54 |
| SHRINKAGE LOSS | 0.00/5.88 | 4.62/11.41 | 4.00/7.23 | 5.30/5.09 | 9.62/10.16 |
| LDS | 0.00/5.88 | 4.62/15.44 | 6.00/10.44 | 9.09/10.48 | 26.45/7.79 |
| KURTOSIS LOSS (OURS) | 5.26/11.76 | 0.00/9.40 | 15.00/26.91 | 21.97/32.04 | 23.76/34.15 |
| PLM (OURS) | 0.00/2.94 | 4.62/11.41 | 5.00/7.23 | 5.30/4.79 | 13.72/4.03 |
| PLW (OURS) | 26.32/8.82 | 7.69/32.89 | 18.00/40.16 | 23.48/39.82 | 6.22/13.22 |

## H ELECTRICITY DATASET STANDARD DEVIATION

Due to space limitations we were not able to report std dev across the 3 runs for the electricity dataset. We present the same in Table 11.

Table 10: Std deviation of results for **Electricity Dataset** (ND/NRMSE/CRPS). All results have been computed across 3 runs with different seeds. Results corresponding to Table 1.

| METHOD | METRIC | MEAN↓ | VaR$_{95}$ ↓ | VaR$_{98}$ ↓ | VaR$_{99}$ ↓ | MAX↓ |
|---|---|---|---|---|---|---|
| BASE MODEL | ND | 0.0023 | 0.0024 | 0.0060 | 0.0258 | 0.1092 |
| | NRMSE | 0.0102 | 0.0033 | 0.0057 | 0.0407 | 0.0919 |
| | CRPS | 2 | 13 | 27 | 69 | 178 |
| CONTRASTIVE LOSS | ND | 0.0075 | 0.0110 | 0.0276 | 0.0425 | 0.5142 |
| | NRMSE | 0.0296 | 0.0108 | 0.0294 | 0.0195 | 0.5382 |
| | CRPS | 17 | 41 | 120 | 184 | 1033 |
| FOCAL LOSS | ND | 0.0018 | 0.0010 | 0.0164 | 0.0203 | 0.1247 |
| | NRMSE | 0.0067 | 0.0009 | 0.0189 | 0.0221 | 0.3053 |
| | CRPS | 4 | 27 | 50 | 68 | 1990 |
| SHRINKAGE LOSS | ND | 0.0021 | 0.0059 | 0.0134 | 0.0248 | 0.3039 |
| | NRMSE | 0.0124 | 0.0048 | 0.0048 | 0.0038 | 0.4650 |
| | CRPS | 5 | 12 | 15 | 170 | 2826 |
| LDS | ND | 0.0014 | 0.0048 | 0.0054 | 0.0401 | 0.7368 |
| | NRMSE | 0.0249 | 0.0074 | 0.0068 | 0.0350 | 0.8518 |
| | CRPS | 5 | 26 | 29 | 81 | 4051 |
| KURTOSIS LOSS (OURS) | ND | 0.0010 | 0.0034 | 0.0084 | 0.0145 | 0.3646 |
| | NRMSE | 0.0153 | 0.0039 | 0.0179 | 0.0170 | 0.4872 |
| | CRPS | 3 | 15 | 44 | 54 | 2845 |
| PLM (OURS) | ND | 0.0021 | 0.0009 | 0.0119 | 0.0308 | 0.3861 |
| | NRMSE | 0.0129 | 0.0010 | 0.0047 | 0.0225 | 0.5220 |
| | CRPS | 3 | 10 | 48 | 83 | 2205 |
| PLW (OURS) | ND | 0.0013 | 0.0026 | 0.0183 | 0.0311 | 0.1256 |
| | NRMSE | 0.0047 | 0.0026 | 0.0164 | 0.0154 | 0.1142 |
| | CRPS | 3 | 8 | 10 | 57 | 1215 |

## I   TRAINING DETAILS

The training procedure employed for the Pareto Losses is as follows:

- Train the base model until convergence
- Fit the Pareto distribution on the loss distribution from the trained model. This is done on the auxiliary loss if one is being used.
- Use the fitted Pareto distribution to implement PLM or PLW and retrain the model.
- The retrained model is the one employing PLM or PLW as per choice.

The training process for Kurtosis loss is straightforward. We use the loss function in Equation (5) directly with one round of training.

## J   ROBUST STATISTICS METHODS

We ran robust regression methods on the task and found that the results do not show improvements on the long tail of error. The methods examined here are Huber Loss and MSLE.

## K   SYNTHETIC DATASETS

We present complete results of our experiments on the synthetic datasets in Table 12. We ran our methods, Kurtosis Loss, and PLM on these datasets as well. Both our methods show significant tail improvements over the base model across all datasets.

Table 11: Results for robust statistics losses on the **Electricity dataset**. Results indicated as **Best**. Huber Loss and MSLE both fail to provide any meaningful improvements on the base model. Moreover, the performance on CRPS is significantly worse illustrating their poor fit for the task.

| METHOD | METRIC | MEAN↓ | VaR$_{95}$ ↓ | VaR$_{98}$ ↓ | VaR$_{99}$ ↓ | MAX↓ |
|---|---|---|---|---|---|---|
| BASE MODEL | ND | 0.0600 | 0.0793 | 0.2251 | 0.4356 | 4.2777 |
| | NRMSE | 0.3069 | **0.0991** | 0.2533 | 0.5430 | 5.5994 |
| | CRPS | 142 | 463 | 1138 | **1996** | 30705 |
| HUBER LOSS | ND | 0.0594 | 0.0822 | 0.2378 | 0.4296 | 3.7959 |
| | NRMSE | 0.2981 | 0.1041 | 0.2492 | 0.5393 | 5.3614 |
| | CRPS | 544 | 1792 | 4892 | 8898 | 31001 |
| MSLE | ND | 0.0608 | 0.0826 | 0.2434 | 0.4336 | 3.9035 |
| | NRMSE | 0.3092 | 0.1162 | 0.2993 | 0.5753 | 5.2328 |
| | CRPS | 601 | 1998 | 5683 | 9935 | 29485 |
| PLM (OURS) | ND | **0.0580** | **0.0791** | **0.2018** | **0.3990** | **3.7827** |
| | NRMSE | **0.2897** | 0.1011 | **0.2396** | **0.4844** | **5.0230** |
| | CRPS | **141** | **449** | **1111** | 2044 | **28992** |

Table 12: Performance on the Synthetic Datasets (ND/NRMSE). Results indicated as Better than DeepAR and **Best** for each dataset.

| METHOD | METRIC | MEAN↓ | $VAR_{95}$ ↓ | $VAR_{98}$ ↓ | $VAR_{99}$ ↓ | MAX↓ |
|---|---|---|---|---|---|---|
| | | | SINE DATASET | | | |
| AUTOREG | ND | 1.2255 | 2.162 | 2.7088 | 2.9306 | 3.1271 |
| | NRMSE | 1.5078 | 2.3134 | 2.7204 | 2.9379 | 3.1271 |
| DEEPAR | ND | 0.0513 | 0.1721 | 0.316 | 0.5913 | 1.5744 |
| | NRMSE | 0.1534 | 0.2009 | 0.3507 | 0.6199 | 1.654 |
| KURTOSIS LOSS | ND | **0.0455** | 0.1412 | **0.2914** | **0.4470** | **1.5571** |
| | NRMSE | **0.1330** | **0.1624** | **0.3455** | **0.5387** | **1.5571** |
| PARETO LOSS MARGIN | ND | **0.0462** | **0.1326** | 0.3014 | 0.7151 | 1.582 |
| | NRMSE | 0.1517 | 0.1563 | 0.3551 | 0.737 | 1.7522 |
| | | | GAUSSIAN DATASET | | | |
| AUTOREG | ND | 0.5730 | 1.0225 | 1.3334 | 1.6226 | 27.6956 |
| | NRMSE | 1.2705 | 1.1212 | 1.4045 | 1.6815 | 39.7474 |
| DEEPAR | ND | 0.4379 | 0.7050 | **0.7908** | 0.8651 | 1.1362 |
| | NRMSE | 0.5518 | **0.8172** | 0.9246 | 0.9908 | 1.3009 |
| KURTOSIS LOSS | ND | **0.4378** | 0.7040 | 0.7973 | **0.8597** | 1.1294 |
| | NRMSE | **0.5518** | 0.8191 | 0.9255 | **0.9865** | 1.2951 |
| PARETO LOSS MARGIN | ND | 0.4391 | **0.7023** | 0.7946 | 0.8674 | **1.1069** |
| | NRMSE | 0.5534 | 0.8194 | **0.9232** | 0.9889 | **1.2786** |
| | | | PARETO DATASET | | | |
| AUTOREG | ND | 1.9377 | 1.1748 | 1.7039 | 2.4782 | 2113.7503 |
| | NRMSE | 81.1652 | 1.4027 | 1.9856 | 2.7312 | 4069.3972 |
| DEEPAR | ND | 0.4416 | 0.8336 | 1.0317 | 1.1763 | 2.015 |
| | NRMSE | 0.6349 | 1.1511 | 1.4295 | 1.6688 | 2.8327 |
| KURTOSIS LOSS | ND | 0.4413 | 0.8345 | **1.0295** | **1.1738** | 2.0326 |
| | NRMSE | 0.6352 | 1.1541 | 1.4305 | **1.6653** | 2.8335 |
| PARETO LOSS MARGIN | ND | **0.4394** | 0.8497 | 1.0473 | 1.1955 | 2.086 |
| | NRMSE | 0.6397 | 1.1694 | 1.4470 | 1.6735 | 2.845 |

