# OpenReview forum: "Taming the Long Tail of Deep Probabilistic Forecasting"
_ICLR.cc/2023/Conference — Submitted to ICLR 2023_

### Official Review · Reviewer_7bpD · 2022-10-25

**Confidence:** 4
**Correctness:** 3
**Technical Novelty And Significance:** 2
**Empirical Novelty And Significance:** 3
**Recommendation:** 5

**Clarity, Quality, Novelty And Reproducibility:**

The writing and statements of this paper are easy to follow. Three new methods Pareto Loss Margin, Pareto Loss Weighted, and Kurtosis Loss are presented for long tail error distribution of probabilistic forecasting. Despite some contributions in identifying and studying the problem, I still have concerns about this paper's motivation and novelty compared to the existing work. The code of the proposed method is not given, which limits reproducibility.

**Strength And Weaknesses:**

This paper analyzes several datasets and identifies a valuable phenomenon: the tail in error differs from the tail in data. This phenomenon reflects that developing algorithms for solving the long tail error distribution in probabilistic forecasting is essential.
Some experiments on real-world and synthetic benchmark datasets show that the proposed method improves the tail performance without cutting the overall performance. In addition, the authors provide a relatively comprehensive analysis of the experimental results and discuss the applicability scenarios of the three proposed methods.
However, my main concerns are (1) the inadequate explanation of the motivation for using Pareto Loss and Kurtosis loss under the long tail error and (2) the experimental results do not stably achieve better on all datasets. Here are the detailed comments:
[Motivation] Although the authors introduce generalized Pareto distribution (GPD) in Chapter 3, the motivations for using GPD and adapting Pareto Loss Margin (PLM) and Pareto Loss Weighted (PLW) are not fully introduced. The paper does not discuss or justify why we would want the specific distribution and not others and why the specific one is being optimized. Similarly, the benefit of using Kurtosis to solve the long tail error is not discussed.
 [Novelty] I think the proposed new method is not innovative enough and does not bring a conceptual breakthrough. As discussed in the paper, using regularization and reweighting is not new; generalized Pareto distribution and Kurtosis are also proposed several years ago and are widely used in many fields. Considering that the authors do not give a new perspective to explain the benefits of using the above methods in the long tail deep probabilistic forecasting, it sounds to me that Pareto and Kurtosis are not used as a regularization or reweighted term in the loss function, so they are combined.
[Experiments] The proposed methods have better average and tail performance on Electricity Dataset compared with the other five baselines and better tail performance on ETH-UCY Dataset. However, the proposed PLW does not achieve better performance on the Traffic Dataset, and the PLW and KURTOSIS LOSS do not consistently guarantee better performance. Based on such experimental results, contributions compared to previous work cannot be significantly proved and requires more discussion. Some parts of the manuscript (abstract, conclusion) seem to overstate the performance and should be reformulated.
[Reproducibility] Although the critical parts of the method are carefully clarified, some implementation for the baselines is listed in the appendix; the paper will be easier to reproduce if the code is given.


**Summary Of The Paper:**

This paper focuses on long tail behavior in deep probabilistic forecasting. Specifically, the authors first observe from the real-world datasets that the tail classes of the data do not necessarily correspond to the tail classes of the error. Then, in order to solve the long tail behavior in prediction error for deep probabilistic forecasting, the authors proposed two new loss augmentation methods: Pareto Loss and Kurtosis Loss. Finally, the experiments on synthetic and real-world datasets indicate that the proposed methods improve the tail performance for some time series forecasting and trajectory forecasting tasks without sacrificing the average performance.

**Summary Of The Review:**

After rebuttal: Thank you for the response. My questions about experiments have been clarified by the percentage improvements in Appendix G. I have updated my score to reflect this.

This paper identifies a critical problem, long tail in forecasting error for deep probabilistic methods, and proposes three new methods to address this problem. However, the novelty and motivation are inadequate, lacking the theoretical analysis of why it is optimal under long tail error distributions.

---

### Official Review · Reviewer_gi62 · 2022-10-26

**Confidence:** 4
**Correctness:** 2
**Technical Novelty And Significance:** 2
**Empirical Novelty And Significance:** 2
**Recommendation:** 3

**Clarity, Quality, Novelty And Reproducibility:**

- Code and data are available, I do not have any concerns about reproducibility from this perspective. However, I have concerns about reproducibility, since probabilities of rare events are analyzed based on only 3 trials. My gut feel is that if I run these 3 trials again, I will not be able to hit the confidence intervals, not even talking about getting the same numbers as reported in the paper.
- Clarity of writing is excellent
- Novelty seems to be moderate as the topic seems to be related a lot to robust statistics and learning, which is a very well developed area


**Strength And Weaknesses:**

STRENGTHS
- Very interesting research topic
- Paper is well-written and easy to follow

WEAKNESSES
- Empirical results do not support the claims in a convincing way
- Empirical results do not seem to focus on the probabilistic forecasting scenario. It is impossible to conclude if the proposed losses lead to better modelling of distributions
- The topic of distributional forecasting is not really handled in the paper and empirical results do not support the probabilistic case put forward in the title. Empirical results mostly analyze large deviations of point errors. Note that CRPS, which has potential to reveal some insights about distributional forecast quality is only reported  in 1 table out of 4. Hence title is misleading.
- Similarly, "taming the long tails..." in the title sounds like a big claim to me. However, empirical results in most cases only provide marginal improvements. Given the empirical results, "taming" sounds like an overstatement to me.
- Using three proposed methods in the tables is misleading. It has the flavor of cherry-picking and using these three methods to create large concentration of bold font in parts of the tables with designator "Ours". I do not think this is a very fair and scientifically rigorous empirical method. Can't we have one method that works all the time?
- It feels that the topic of the paper is very close to robust statistics, Huber loss and other robust losses. However literature review does not cover any literature on robust estimation. I would think this is a very important topic to cover in this paper.
- None of the robust losses are used as baselines. At the same time, contrastive loss is used as a baseline. Is contrastive loss really relevant for better predicting outliers? I would rather focus on providing some baselines based on robust losses, since these are much conceptually closer.
- "The idea behind our Pareto Loss is to fit the GPD pdf in equation 2 to the loss distribution and use it to increase the emphasis placed on the tail samples during training". I am not sure I follow the logic here. I am under the impression that using a heavy-tailed PDF in NLL actually reduces the NLL of these samples from the tails by modulus. So their magnitude reduces relative to the other samples. Hence, I would actually expect their effect in the overall loss to diminish. Can you backup this claim with math or empirical examples?
- The example Figure 3 makes little sense to me. I don't think that DeepAR is particularly strong in point estimation and it's obviously not a suitable model to handle sinusoids, that's why the sinusoidal case produces those errors. In general, any model with insufficient capacity and unsuitable for the task will have a tendency to produce heavy tailed errors. However, I do not think that in this case the solution should be to modify a loss. The solution should be to find an appropriate model for the problem. The only circumstance under which the authors' logic will make sense to me is if we really want to use DeepAR to solve all the problems in the world. However, I do not think this is very practical. Therefore, I am not following the motivation here and I suggest that Section 4.2 has to be removed or rewritten to provide a good motivation.
- I am not sure that using only two models in conjunction with proposed losses is enough to prove that the losses are universally applicable. I would much preferred if there was one proposed loss that was empirically shown to work well with 5 different forecasting models, which to me would be a solid proof of a working loss.

**Summary Of The Paper:**

This paper addresses the problem of probabilistic forecasting and proposes new loss functions that are supposed to reduce the probability of catastrophic errors.

**Summary Of The Review:**

Although the topic of the paper seems interesting and directionally, I would definitely like to see more papers handling probabilistic forecasting in statistically consistent way, I think this paper fails in a few key respects. Therefore, I recommend reject. The key respects are as follows:
- Probabilistic forecasting is not handled, no empirical backup for the proposed losses to result in better distributional forecasting
- Empirical framework seems to be flawed and empirical results are not convincing

---

### Official Review · Reviewer_953o · 2022-10-28

**Confidence:** 5
**Correctness:** 3
**Technical Novelty And Significance:** 3
**Empirical Novelty And Significance:** 3
**Recommendation:** 6

**Clarity, Quality, Novelty And Reproducibility:**

Training process not clear see above.


Sentence: "Makansi et al. (2021) observed similar long-tail error in trajectory and proposed to use
Kalman filter prediction performance to measure sample difficulty. However, Kalman filter is a
different model class and its difficulties do not translate to deep neural networks used for forecasting."

is unclear, since it assumes that the reader knows that Makansi is using the KF as a component of their approach.

**Strength And Weaknesses:**

Strength
- Interesting idea with good motivation with respect to existing similar approach (Makansi 2021)
- Application shown in different domain, thus indicating the result may be useful to multiple fields

Weaknesses
- training method not clearly described. Eq(3-5) describe reweighting or margin rescaling to change model behavior. Is (2) fitted at every epoch? is it fit once after a first training iteration (multiple data sweeps)?

**Summary Of The Paper:**

This paper addresses the issue of long tailed distribution of errors in forecasting problems. While in general in forecasting problems the main focus is to improve samples that lies around the average of the error distribution, in this work the attention is put on the long tail of the error. Differently from Makansi 2021, where the difficulty of the sample is computed using the error of a Kalman Filter, here a tail sensitive loss is introduced. The main observation to motivate this approach is that the KF error may not correlate with the error a deep probabilistic forecasting model is attaining. Therefore, it is not correct to use such error to re-weight the samples.

In this work two losses are introduced: one derived by fitting model error with Pareto distribution; and one based on the Kurtosis of the error distribution. The main idea is to use these losses to give more weight to samples that are on the long tail of error.

Experimental results show that with respect to baseline models there is some improvement

**Summary Of The Review:**

The paper addresses an important problem. The issue addressed is known but the proposed solution is novel. Results are somehow showing that the approach may give some benefit. Training procedure is not clearly reported.

---

### Official Review · Reviewer_AHB8 · 2022-10-31

**Confidence:** 4
**Correctness:** 3
**Technical Novelty And Significance:** 2
**Empirical Novelty And Significance:** 3
**Recommendation:** 5

**Clarity, Quality, Novelty And Reproducibility:**

The paper was in general very clear and the philosophical tone quite pleasant to read!

On reproducibility: The paper doesn't explain how the GPD distribution is exactly fit to the data. A reference or some discussion of the procedure in the appendix would have helped. Also, it is unclear what steps were taken to tune the hyper parameters of the competing methods that were tested. Was a similar grid search procedure used as was used to train the hyper parameters of the approach proposed in this paper?

Novelty seems weak since the existing methods as described in the paper also look at the loss of one round of training to inform the second round. However, this paper attempts to explicitly model the loss of the first round of training which does appear to be slightly novel.
The stark improvement in results here does suggest that something interesting might be going on, but it is not clear to why.



**Strength And Weaknesses:**

As mentioned, the paper provides a number of good philosophical points include the distinction between long-tailed data distributions and long-tailed error distributions. There is also a very good discussion of related work in the literature.

The other strength of the paper is that the technical results seem quite interesting and do indicate a strong across-the-board improvement. The examples in Figure 4 appear extraordinarily good!

The three proposed methods sound quite simplistic and it is hard to immediately see why they would provide an improvement over some of the other methods for reweighing and regularizing the loss. Indeed the paper never discusses precisely why the proposed loss modification techniques work better than others. There is a brief discussion at the end of page 7 claiming that the improvement was likely due to bounded weights, but there is nothing tangible to explain the improvement.

I would have liked to see some amount of ablation studies to demonstrate that the methods chosen in this paper were carefully considered among other similar choices.

- The paper revolves around the Generalized Pareto Distribution (GPD) which certainly is very relevant modeling tails. However, a natural question that does arise is whether we could have used a Cauchy distribution which is also very good at modeling long-tailed behavior.

- Similarly, for the loss on equation 5 instead of a using the fourth moment couldn't we have used the second moment? How would that have changed the results.

- The paper describes doing two rounds of model training with the losses of the first round helping determine the loss function used in the second round. Can this procedure be repeated for a third or a fourth round to get even better results? Is two rounds the optimal number?

The paper mentions that in on page 3 that in two runs of their DeepAR model training on the Electricity forecasting dataset there were only 3.5% tail error samples in common. This begs the question of whether we could have simply trained two models and averaged their predictions to get better overall predictions?

The paper never goes into why only sequence modeling tasks were used for evaluation.

Minor:
- When the author use the phrase "exhibits long tail behavior" is there a formal definition of what this means?
- The conclusion mentions "more principled ways to tune hyper parameters" but I didn't see in the main text how these were tuned in the first place. It would be good to add a sentence or two in the experiment section.

**Summary Of The Paper:**

The paper provides a few techniques to modify the loss function to be used by a learner to ensure that the learned model doesn't exhibit extreme loss values on test examples (using the original loss function).

The paper provides a good overview of related work in this area and makes two important philosophical points worth considering. 1) It is impossible to identify a fixed set of tail samples. Different learning methods and in fact different runs of the same learning method could produce extreme loss values on completely different samples. 2) "Overemphasizing tail examples might hamper the learning for other samples." In other words it is important to limit the amount of extra weight given to samples with extreme loss values.

**Summary Of The Review:**

Very interesting results using a simple method which seem hard to explain easily.

---

### Decision · Program_Chairs · 2023-01-20

**Decision:**

Reject

**Justification For Why Not Higher Score:**

Although one reviewer votes for acceptance, the support is very weak. In fact, that review is a bit superficial.


**Justification For Why Not Lower Score:**

N/A


**Metareview: Summary, Strengths And Weaknesses:**

While various methods have been proposed for long-tailed data distributions, this paper aims to arouse the awareness of the research community on some properties related to long-tailed error distributions instead. The paper motivates well the differences between the two and also their implications. The strong experiment results are also encouraging. However, the proposed Pareto loss (based on generalized Pareto distribution) and Kurtosis loss are not new. This would probably be fine if the paper made contributions in other aspects. For example, despite their simplicity, why do they work so well in the experiments? Unfortunately, an in-depth deliberation of this important aspect is lacking, let alone a scientific treatment. While this work has potential for potential for publication in the future, the current paper requires substantial revision in order to address the concerns. We hope our comments and suggestions can help the authors in revising their paper.